# Understanding the Opportunities to Mitigate Carryover of Imidazolinone Herbicides in Lowland Rice

**Vinicios Rafael Gehrke** [ID]**, Marcus Vinicius Fipke, Luis Antonio de Avila** *[ID] **and Edinalvo Rabaioli Camargo** *[ID]

Crop Protection Graduate Program (Programa de pós Graduação em Fitossanidade), Federal University of Pelotas (Universidade Federal de Pelotas), Pelotas 96160-000, RS, Brazil; viniciosraffael@hotmail.com (V.R.G.); marfipke@gmail.com (M.V.F.)
* Correspondence: luis.avila@ufpel.edu.br (L.A.d.A.); edinalvo.camargo@ufpel.edu.br (E.R.C.)

**Abstract:** (1) Background: The Clearfield™ system (CL) is currently the primary tool for selective weedy-rice management in irrigated rice. However, herbicide persistence in the soil may cause damage to successive crops. Thus, it is necessary to understand agricultural practices that can favor the dissipation of these herbicides. The objective of this study was to analyze the factors that affect the persistence of imidazolinones and to use this information to provide management strategies to mitigate carryover in lowland rice. (2) Methods: A literature review was performed, and the publications were selected using the soil half-life parameter. The data were summarized according to the biotic conditions, soil parameters, and environmental variables. (3) Results: Imidazolinone dissipation in soil occurs primarily through biodegradation. The herbicide biodegradation rate depends on environmental conditions such as temperature and bioavailability of the herbicide in the soil. Herbicide bioavailability is affected by soil conditions, with higher bioavailability in soil with higher pH, less clayey texture, moderate organic matter content, and higher soil moisture levels. Therefore, environmental conditions that favor biological activity, especially high temperatures, reduce the herbicide half-life in the soil. Strategies to mitigate carryover should focus on improving herbicide availability and enhancing biological activity in the soil, especially in the rice off-season, when low temperatures limit herbicide biodegradation. Cover and rotational crops, such as ryegrass and soybean, are recommended, with the potential to mitigate soil residues. (4) Conclusions: The establishment of crops other than rice would automatically enhance degradation rates as soil amendment practices such as pH correction and drainage practices would favor soil availability and biological activity.

**Keywords:** persistence; clearfield; lowland; half-life

## 1. Introduction

To meet food demand, global agriculture is inextricably linked to pesticide use due to its practical and affordable nature and its importance in increasing food security. However, the pesticide fate in the environment could result in various adverse effects, including damage to non-target organisms, groundwater contamination, and persistence in the water, air, plants, animals, and soil [1].

The persistence of herbicides, particularly pre-emergence herbicides, in the soil is a critical factor in their effectiveness and the potential carryover effect on successive crops. While herbicide residues in the soil are necessary to control weeds during the life cycle of the crop of interest, persistence beyond the growing season can be problematic since the herbicide can harm subsequent crops if present at high enough concentrations.

The imidazolinone herbicides (imazapyr, imazapic, imazethapyr, imazamox, and imazaquin) are, in general, soil-persistent herbicides and can carryover, affecting rotational crops. These compounds act by inhibiting the acetolactate synthase (ALS) enzyme, vital in the biosynthesis of branched-chain amino acids [2]. In general, these herbicides are characterized by a slow degradation rate in the soil and specific selectivity [3,4]. An

example is imazapyr, which is used is primarily in non-agricultural areas. On the other hand, imazethapyr and imazapic were indicated for certain crops such as beans and soybean in highland cultivation systems where sufficient degradation occurs during the growing season so that damage is not observed in subsequent crops [5–7].

However, this practice changed with the introduction of Clearfield™ (CL) system, which includes the use of imidazolinone-resistant cultivars. The first cultivars launched had limited selectivity and were called first-generation cultivars. A new mutation was obtained later and produced more resistant genotypes called the second-generation cultivars. The first-generation cultivars were tolerant to imazethapyr or the mixture of imazethapyr + imazapic. The second-generation included the mixture of imazapyr + imazapic, which allowed these herbicides to be used in a new environment, i.e., lowland irrigated rice cultivation [8,9]. Herbicide dynamics in these environments are remarkably different from those in uplands, mainly due to the maintenance of standing water during the growing season, which alters transport processes (e.g., leaching, runoff, and degradation).

Environmental conditions in the lowlands negatively affect herbicide degradation processes, prolonging imidazolinone persistence in the soil that can affect non-tolerant crops in both the winter and summer (carryover). There are reports of imidazolinone carryover to ryegrass, soybean, sorghum, corn, and conventional rice cultivated following CL rice [10–12]. As a consequence, there is a greater reliance on CL rice monoculture. There is, therefore, a need to evaluate possible management approaches that will lead to greater degradation of imidazolinones in the soil.

A herbicide's half-life ($t_{1/2}$) is the best estimate of herbicide persistence in the soil and is a valuable parameter to compare herbicides. The herbicide half-life coefficient is the time required for the herbicide concentration to be reduced by 50% of its initial concentration [13]. Several half-life values have been reported for imidazolinones in the literature, with values ranging between 10 and 300 days, depending on the conditions under which these compounds are applied [14].

This work aims to conduct a systematic review of the reported half-life of imidazolinones in the soil and analyze and group these values to support the development of management strategies to reduce imidazolinone persistence in the soil after its use in CL rice crops.

## 2. Materials and Methods

This systematic review was carried out using the Prisma Protocol. Literature searches in Portuguese and English using the *Web of Science*, *Scielo*, *Science Direct*, and *Scopus* databases and the search string "*(Degradation OR half-life OR degradação OR meia-vida) AND (imidazolinone OR imidazolinona OR Imazapyr OR Imazapir OR Imazapic OR imazethapyr OR imazetapir OR imazaquin OR imazamox)*"; records were *downloaded* in *\*.ris* format. The manuscripts were obtained from the databases cited between 10 February 2020 until 30 March 2020. The data obtained from this search were processed using the R software *v4.0.1* [15]. First, the fuzzy logic algorithm of the *revtools v0.4.1* package [16] was used to remove duplicates (Parameters—function: fuzzydist; method: M Ratio; maximum distance: 0.1). This process was performed in three steps using the title, DOI, and the abstract as criteria for defining duplicates. Then, the *tm* package [17] was used to build a corpus object, in which selection filters were used to remove spaces, numbers, terms with less than three letters, and the stemming filter, which standardizes different inflections of the same term. The same package was used to construct a document-term matrix, which was processed using the Latent Dirichlet Allocation (LDA) topic model. A matrix was created with the frequency of terms in the title, abstract, and keywords for each document, and the $\beta$ index (probability of a term being assigned to a topic) and $\gamma$ index (likelihood of a document being assigned to a subject) were calculated [18,19]. The $\gamma$ index was then used to plot an ordinary axis graph of the topics created and select works based on the titles and abstracts. Later, a new term matrix was created with the selected works, and the frequency of the terms was calculated. A word cloud was prepared using the *wordcloud* package [20], with

each word's size varying according to its frequency in the selected records. Finally, the data of the articles were downloaded and processed with the *metagear* package [21]. The articles were selected using the determination of the half-life of imidazolinones in the soil in first-order dissipation models.

The data were extracted using categorical variables (soil texture, herbicide, enantiomers, the methodology used, soil temperature, and organic matter) and quantitative factors (soil pH and soil moisture). Data related to statistical analysis were also extracted, such as standard deviation, standard error, confidence interval, and correlation coefficient, for further meta-analysis.

## 3. Results

The results from database searches and the recorded selection are presented in Figure 1. The search criteria resulted in 1063 records, while an additional record that was not included in the search results was incorporated during the data screening process. The fuzzy algorithm allowed for identifying 228 duplicate records, and the remaining 835 records were grouped into ten topics (stipulated value) using the LDA model.

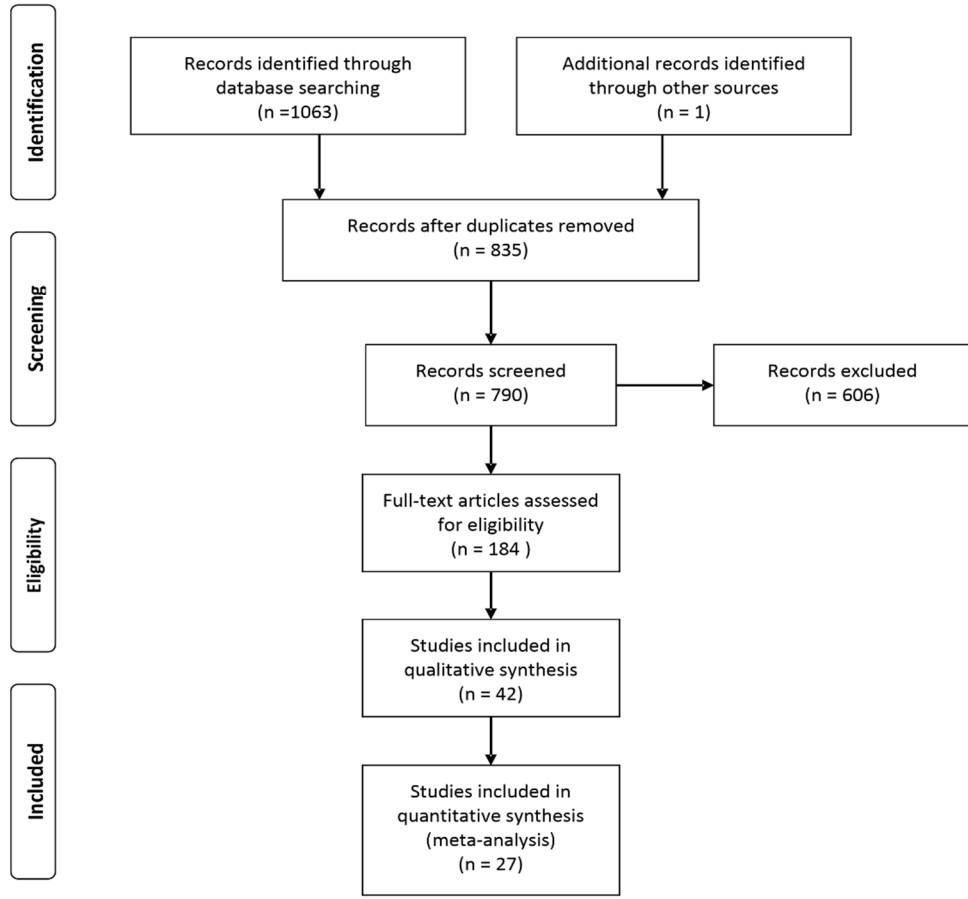

**Figure 1.** Flowchart for the selection and inclusion of records using the PRISMA protocol [22].

The results of the topic model can be seen in Supplementary File (Supplementary Figure S1). They can be arbitrarily classified as environmental dynamics (1, 7, and 8), weed management (4 and 10), quantification methodologies (3), omics (5 and 9), and generic (2 and 6). A total of 184 studies were selected for full-text analysis using the inclusion/exclusion criteria. The relationship between the selected records and their topics can be found in the Supplementary File (Supplementary Figure S2).

The word cloud of the selected records, which was prepared with the LDA topic model (Figure 2), showed that the terms soil, "herbicid", "degrad", imazethapyr, imazapyr,

and imazaquin were the most frequently used in the selection. Their relative frequency in the records is shown as differences in their size and color in the word cloud. These terms' frequency demonstrates that the selection of abstracts included collecting data relevant to the processes underlying imidazolinone soil degradation processes and the factors involved in herbicide degradation (smaller, gray terms in the word cloud).

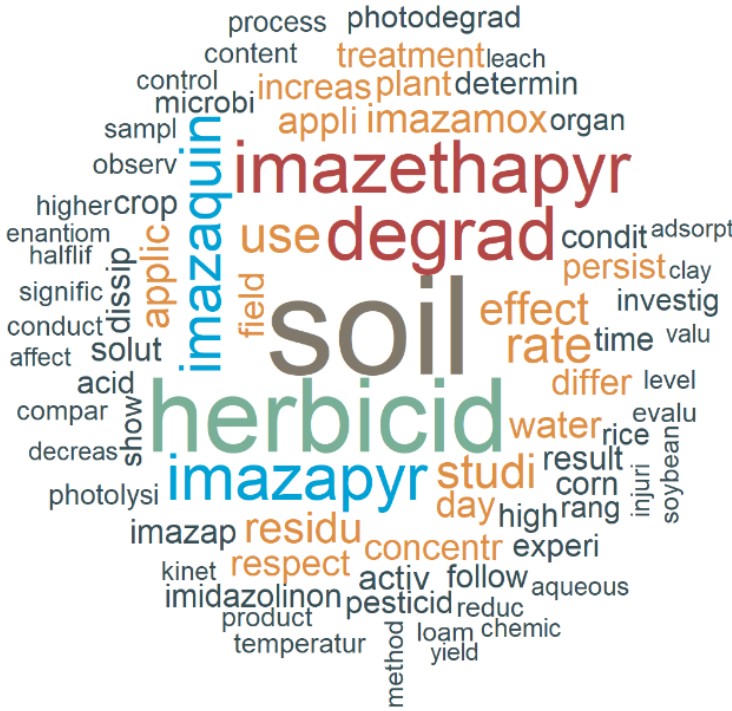

**Figure 2.** Word cloud prepared using the frequency of terms found in the records selected to obtain imidazolinones' half-life. The size and color of the terms denote their frequency in the abstracts of the documents.

Of the records selected for full-text analysis, 20 were not available online and were subsequently excluded. A total of 606 of the complete studies analyzed did not meet the previously established requirements, meaning that 27 papers were used in this study.

Initially, the intended methodology to process half-life data was a meta-analysis. However, many of the data could not be included due to a lack of information on the variance in half-life coefficients [23]. A second limiting factor is intrinsic to the method of the half-life determination itself, which is determined by Equation (1)

$$t_{\frac{1}{2}} = \frac{lnln\ 2}{k} \tag{1}$$

where *k* is the estimated angular coefficient of the linear regression (*β*) of the herbicide concentration, which is time-dependent. Thus, there is a limiting factor in determining the variation in the dataset, especially in calculating the effect size, which is necessary to evaluate each record's individual effect on analyzing the overall results [24]. Thus, a more descriptive approach was used to categorize and assess the half-life values observed in the selected records.

First, a distribution histogram was prepared (Figure 3), highlighting each record's values by assigning a color palette to the selected records employed throughout this study (Supplementary Figure S3). The histogram shows that most of the studies report a half-life of imidazolinones between ten and 100 days, though some authors report values longer than 600 days. Further analysis of the extreme outliers (>600 days) indicated that these were values from studies with contrasting biotic conditions [25–28].

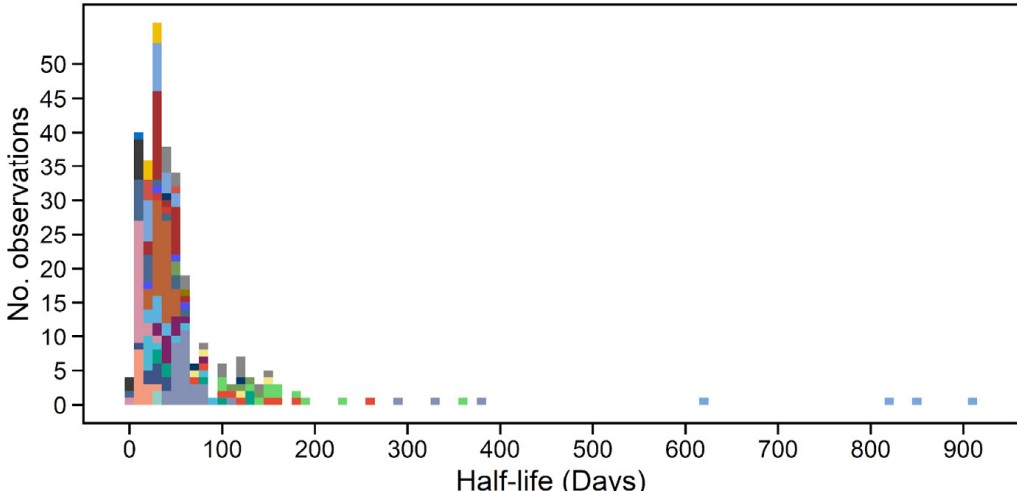

**Figure 3.** Histogram of the distribution of the imidazolinone half-life values in the selected search records. Each record's values by assigning a color palette to the selected records employed throughout this study which was included on Supplementary Figure S3.

Records were therefore grouped on whether they measured the half-life in sterile or non-sterilized soils. The results in Figure 4 show a considerably longer half-life under sterile conditions, underscoring the leading role of biotic processes in imidazolinone degradation [25–27]. Imidazolinone herbicide can be degraded by abiotic means as well, as these herbicides are sensitive to photodegradation in water, depending on its turbidity and soluble organic matter content (indirect photolysis) [29–31]. However, photodegradation in the soil is limited to the topsoil (0.2–0.7 mm) and is restricted to less soluble compounds such as imazethapyr and imazaquin, as the others tend to move more easily to deeper layers of the soil [30,32,33].

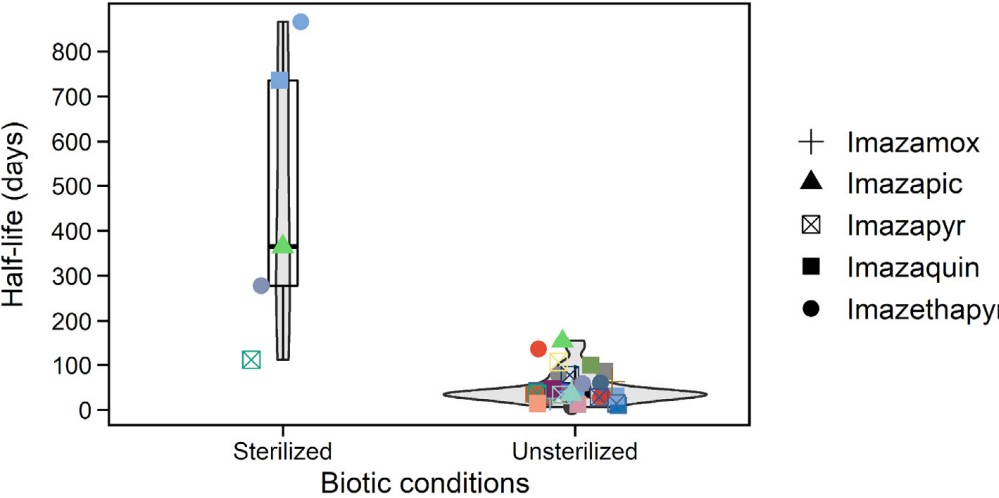

**Figure 4.** Half-life of imidazolinones in sterile and unsterilized soil.

Biodegradation is the main factor to be considered in the reduction of imidazolinone residues. This primarily requires the availability of the herbicide in the soil solution, which depends on texture, pH, soil organic matter content [34,35], and environmental factors such as moisture and temperature [36].

To understand the effect of soil texture on the half-life of imidazolinones, only the half-life values reported in works that evaluated different soil types were examined to isolate the other variables. The values examined were plotted on ternary plots allocating their respective texture classifications to the USDA soil texture triangle (Figure 5). Figure 5

indicates the soil trend with a higher clay content to degrade imidazolinones at a slower rate [37–39] due to the herbicide sorption to the soil clay constituents, especially iron oxides [34,40].

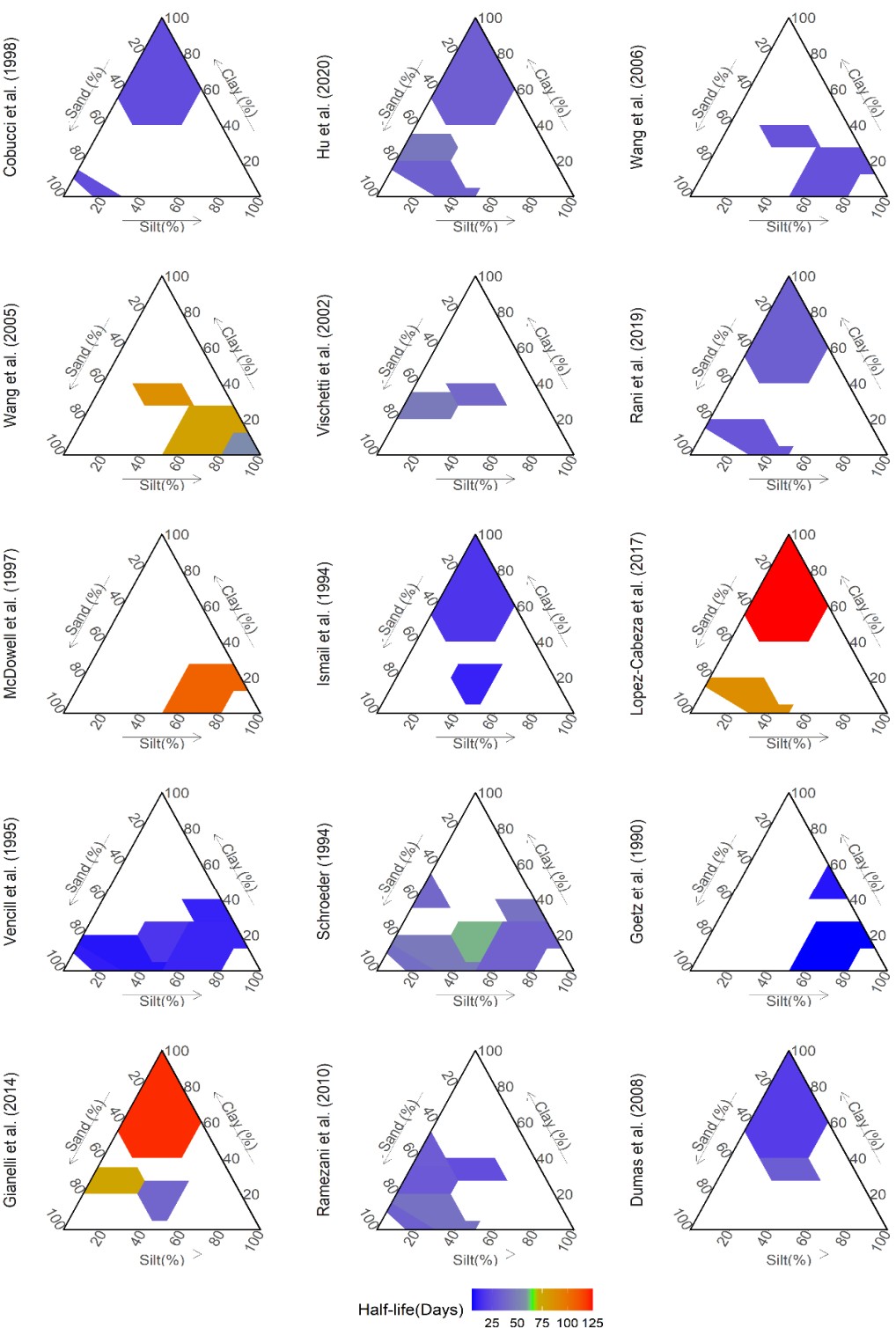

**Figure 5.** Half-life of imidazolinones as affected by soil texture (USDA Soil Texture Triangle, Source: Hamilton and Ferry [41]).

In addition to soil texture, the organic matter content can also determine herbicide availability [42]. However, there is no specific relationship between the organic mat-

ter content and imidazolinone degradation based on this dataset, as shown in Figure 6. Since a higher organic matter content tends to favor microbiological activity in the soil, Wang, et al. [43] evaluated its effect by incorporating 10% cattle manure into the soil, which reduced the half-life of imazaquin from 21 to nine days.

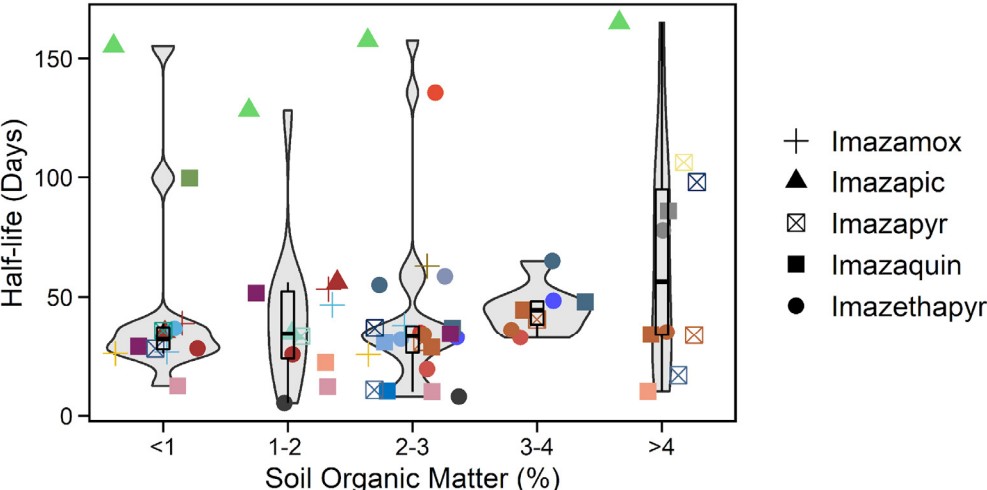

**Figure 6.** Half-life of imidazolinones (days) at different soil organic matter contents (%).

The herbicide dissipation in the soil is dependent on the organic matter content, as reported by Su, Hao, Ding, Wu, Xu, Xue, Shen, Sun, and Lu [25], who assessed the degradation of imazapic as a function of increasing organic matter in amended soil with poultry bedding and found that soil organic matter levels of 0.55% and 1% sped up the degradation rates, while organic matter levels of 2.5% and 4% slowed it down. An excessive increase in soil organic matter reduces the degradation rate due to greater sorption of the herbicide [44].

Furthermore, due to imidazolinone sorption by a hydrophobic partition, the effect of soil organic matter tends to be significant only under low pH conditions [37,45]. Soil pH has a complex impact on imidazolinone availability. The imidazolinones are weak acids, meaning that the carboxyl group's protonation causes the soil's hydrophobic fraction to attract the herbicides in low pH soils. Increased pH leads to the carboxyl group's deprotonation, making the herbicide an anion, repelled by the negative charges of clays and organic matter, resulting in a complex interaction between the herbicide and the components of the colloidal fraction [31,34].

Imidazolinone degradation rates were not directly related to soil pH across studies that examined different textures, as can be seen from the trend lines in Figure 7. However, when the increase in soil pH was isolated, as in the study of Su, Hao, Ding, Wu, Xu, Xue, Shen, Sun, and Lu [25], shown as a dotted line in Figure 7, the reduction in the half-life of imazapic as pH increases is noticeable. Aichele and Penner [36] compared soils with pH 7 that were acidified to pH 5 and noted a reduction in the degradation rates of imazaquin, imazamox, and imazethapyr, indicating lower metabolization of these herbicides at lower pH values. According to some authors, the higher sorption of imidazolinones at pH levels below 6 is the main pH-related cause of lower degradation rates of these herbicides [36,46].

Furthermore, the increase in pH provides a more favorable environment for microbiota development, mainly bacteria, responsible for degrading xenobiotics in the soil. Soil liming to raise soil pH increases microbiological activity and the degradation rates of other pesticides such as glyphosate [47], chlorsulfuron [48], and fenamiphos [49]. Singh, Walker, Morgan, and Wright [49] further noted that soils with neutral or alkaline pH are associated with greater microbial population stability, enabling the microbiota to quickly adapt to degrade the same compound with subsequent applications. The greater availability of some nutrients at higher pH values is an additional factor that promotes the development

and increases the production of catalytic enzymes in the soil [50], which may assist in the degradation of imidazolinones.

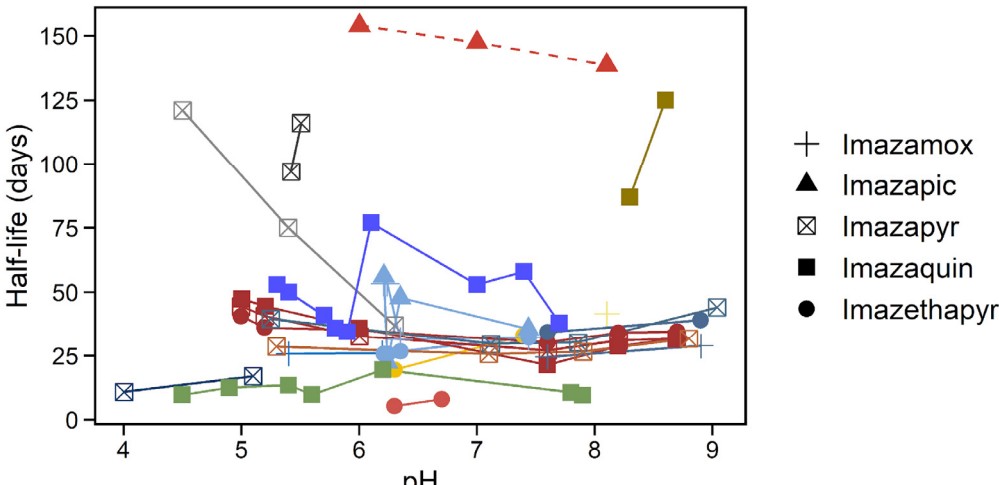

**Figure 7.** Relationship between soil pH and imidazolinone half-life (days). The dashed line corresponds to the work of Su et al. [25].

As previously mentioned, the availability of the herbicide in the soil solution is a requirement for the dissipation process. Therefore, soil moisture is a critical factor in imidazolinone degradation as water acts as a solvent to make the herbicide available in the soil solution. In this study, soil moisture was analyzed in the dataset using two parameters, the soil-to-water ratio (mass/mass; Figure 8a) and the percentage field capacity (Figure 8b). This was due to the lack of standardization and information in the selected records.

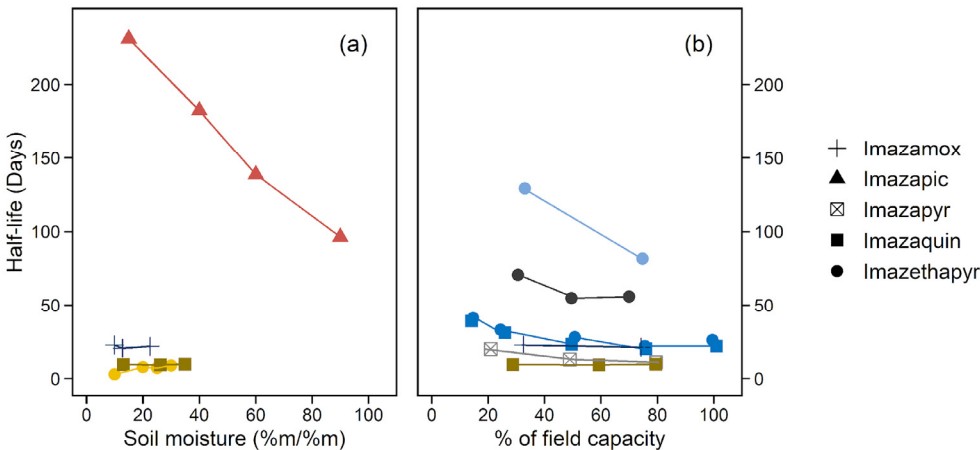

**Figure 8.** Relationship between imidazolinone half-life (days) and soil moisture (%m/%m) (**a**), and percentage field capacity (**b**).

In general, imidazolinone degradation is slower under conditions of lower soil moisture. As highlighted by Wu, He, Dong, Zhou, and Zhang [27], the average half-life was 15 days longer in soil with moisture levels of 30% field capacity than in soils with 50% and 70% field capacity. These results are consistent with findings by Ismail and Ahmad [51], who also noted that moisture-related reductions in the half-life were greater in soils with clayey textures.

On the other hand, Flint and Witt [28], who assessed the degradation rate of imazaquin and imazethapyr at moisture levels of 15%, 25%, 50%, 75%, and 100% field capacity, noted a gradual increase in the degradation rate up to 75% and a reduction at 100% field capacity, especially for imazethapyr. These results highlighted a gap in the literature considering the

degradation of imidazolinones under higher moisture or soil saturation conditions, as in rice-producing areas, flooding results in a hypoxic environment [52].

As a rule, the biodegradation of herbicides tends to be slower in an anaerobic environment, mainly due to the low efficiency of anaerobic metabolic pathways. However, Wang, et al. [53] assessed imazapyr degradation under aerobic and anaerobic conditions in four soils from China. The degradation rate was lower in most soils under anaerobic conditions (13.6, 31.5, 19.5, and 22.3 days) than in the same soils under aerobic conditions (39.6, 25.9, 44.1, and 29.7 days), with specific soil organisms playing a key role. The same work examined metabolites formed in both pathways and found that the possible degradation mechanism occurs through the demethylation–hydroxylation of the imidazole ring, which can happen in the other compounds that contain the same structure, similar to that proposed for the anaerobic degradation of imazosulfuron (Sulfonylurea) proposed by Morrica, et al. [54].

The degradation of imidazolinone herbicides in flooded soil has not yet been explored in detail in the literature since these conditions are unique to rice cultivation. Results reported for other compounds demonstrate a case-specific relationship between each molecule's physicochemical characteristics and the soil, microorganism populations, and the environment. Some herbicides may degrade further under anaerobic conditions due to increased availability in the soil solution. In contrast, others may have reduced degradation due to the restriction of aerobic microorganisms [55,56].

Heiser [57] reported that the degradation rate of imazamox, imazethapyr, and imazapic was higher in flooded soils than under non-flooded conditions, attributing these results to the presence of sufficient oxygen concentrations in the topsoil to maintain the metabolism of aerobic microorganisms. Junkes, et al. [58] evaluated the dissipation of imazapyr in a continuous and intermittent irrigation system, observing half-lives of 182 and 42 days, respectively. The above studies all concluded that high soil moisture or soil saturation increases herbicide availability in the presence of a certain water level, allowing degradation to occur via aerobic metabolism, which is most prominent in intermittent systems that will enable both aerobic and anaerobic degradation to occur.

Some authors note that microbial degradation is affected primarily by soil moisture and secondly by temperature [25,28]. Temperature is directly related to the growth rate, microorganism metabolism, and enzymatic kinetics [59]. The dataset analyzed in this study supports the relationship between imidazolinone degradation and temperature, as shown in Figure 9. Temperature increases result in a reduction in the half-life. The optimum temperature appears to be between 25 °C and 35 °C [25,60]. Meanwhile, temperatures below 20 °C increase the persistence of herbicides in the soil [28,61].

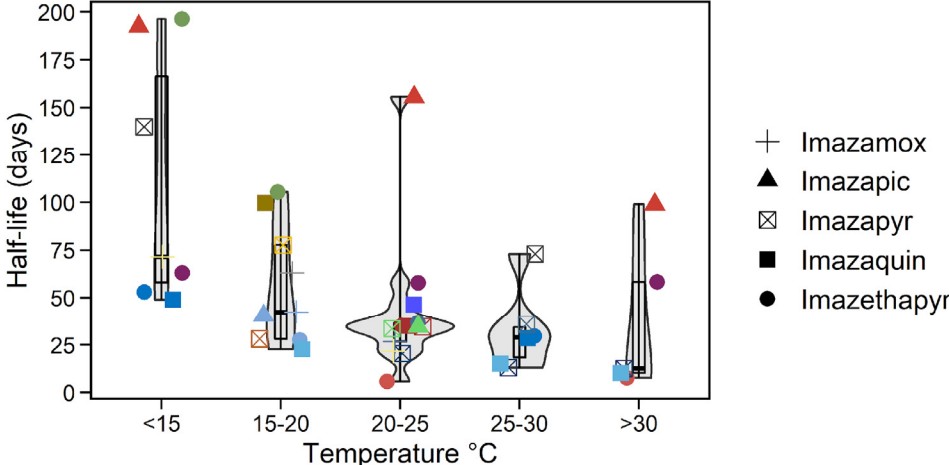

**Figure 9.** Effect of temperature on the half-life of imidazolinones (in days).

Another aspect that may affect degradation is herbicide enantiomers' use to replace the racemic mixture [62–64]. Enantiomers are variations of a molecule when an asymmetric carbon is present, meaning one with four different bonds. In imidazolinones, optical isomers of the methyl and propyl radicals in the 4-carbon of the imidazole ring are possible, allowing for *R*- and *S*-isomers. The physical and chemical characteristics of the isomers are not different. However, biological enantioselectivity can occur since plant enzymes can have a greater or lower affinity for one of the enantiomers [63].

Due to its higher affinity with the ALS enzyme, the *R*- enantiomer of imidazolinones causes greater ALS inhibition than the *S*- enantiomer of the racemic mixture [63,64]. The same behavior is reflected in imidazolinones' biodegradation since it occurs through enzymatic catalysis in the soil. In addition to its higher agronomic effectiveness, some authors point out that the *R*-isomer has a slightly shorter half-life in the soil, as shown in Figure 10. However, its half-life remains sensitive to edaphoclimatic variables and is dependent on the particular microorganism community of each soil [27]. However, the potential use of enantiomers is not only limited by the lack of information about their herbicidal activity and environmental behavior but rather because no commercially available imidazolinone herbicides are sold with enrichment of the *R*-enantiomer. Only the racemic mixture is sold, mainly due to the difficulty and cost of producing and isolating isomers on a large scale [65].

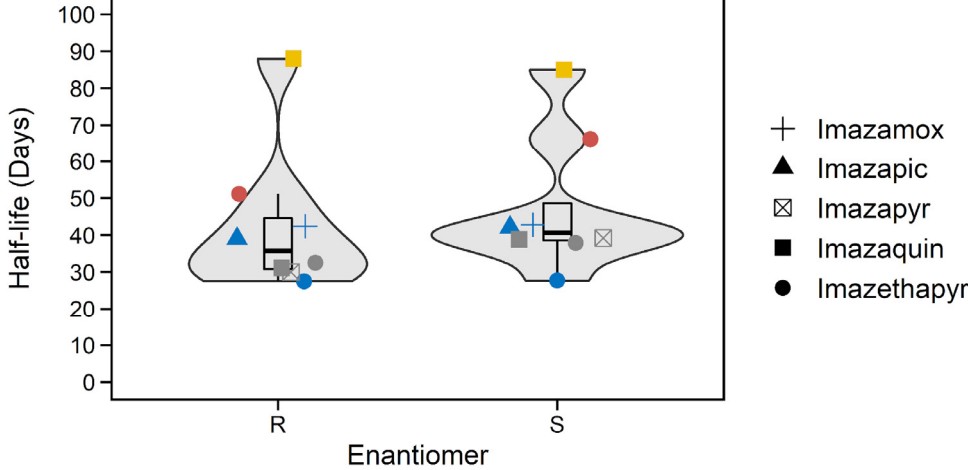

**Figure 10.** Half-life (days) of the *S*- and *R*-enantiomers of imidazolinones.

This discussion of the available data clarifies that several edaphoclimatic factors act on the degradation rate of imidazolinones. In general, conditions that favor aerobic microorganisms lead to higher rates of degradation of imidazolinones. As observed, situations of good organic matter in the soil (3%), neutral pH range (6–7), soil moisture from around 75% field capacity up to saturation, and a temperature range between 25–30 °C are the most suitable conditions for the development of aerobic microorganisms and, consequently, their combination would favor the rate of degradation of imidazolinones. These patterns may be of interest when addressing gaps in management practices to reduce post-harvest herbicide residue, as discussed in the following section.

## 4. Management of Imidazolinone Residues in Lowlands

No records on imidazolinone degradation in lowland soil were found, as existing studies limit their focus to the surface water degradation rate [66–68]. Lowland soils drain poorly due to characteristics such as flat topography, the presence of sub-surface clay horizons with low hydraulic conductivity, and the proximity to the water table that makes it challenging to drain these soils. While these features favor rice cultivation, they also prevent the degradation of imidazolinones.

According to the recommendations, these herbicides can be applied before emergence or just after emergence, followed by a second application at the $V_3$–$V_4$ rice stage, before or simultaneously with flooding [69]. Rice soil is flooded for three to four months after application, during which aerobic microorganisms that can break these herbicides down are suppressed. After the rice-growing season, the residual herbicide concentration can result in carryover effects being detrimental for the subsequent crops [11], such as winter crops, soybean, and non-tolerant rice, with effects lasting for up to two years [12,70–72].

Furthermore, the degradation rate between growing seasons may be reduced by low temperatures since the period in which surface water is drained for harvesting coincides with a decline in temperature (Figure 11). The relationship between temperature and the degradation rate was discussed previously, but it is worth mentioning that aerobic microorganisms' metabolism is limited by temperature. Therefore, other factors should be altered where possible to favor aerobic microbial activity, which will improve the degradation rate.

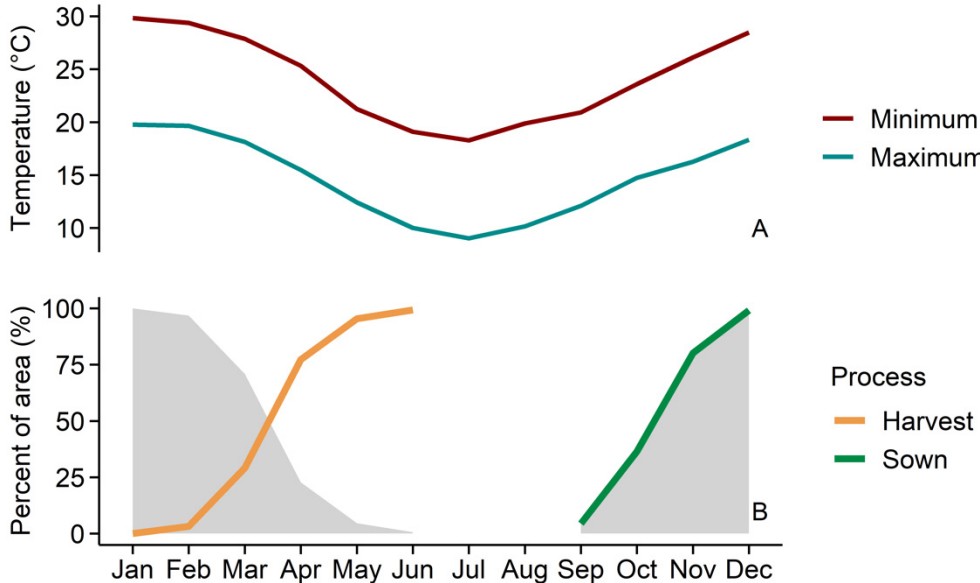

**Figure 11.** (**A**) Monthly average minimum and maximum temperatures from 2000 to 2019 (Source: Dados da Rede do INMET [73]), (**B**) Average harvest and planting activity in the Rio Grande do Sul state, Brazil between 2010 and 2019 (Source: IRGA [74]). The shaded area represents the percentage of Rio Grande do Sul State (largest rice producer in Brazil) acreage with rice crop progression from initial establishment (sowing starting in Sep.) to final harvesting (end of May).

Firstly, post-harvesting drainage should be carried out to promote soil aeration and provide soil preparation conditions since flooding reduces microgalleries in the soil [52]. Along similar lines, Kraemer, et al. [75], assessing the effect of soil preparation during fallow periods on the carryover of a herbicide mixture of imazethapyr and imazapic to non-tolerant rice, found out that the initial injury to the non-tolerant cultivar IRGA 417 was lower when the soil was prepared 2–4 times between May and October compared with preparation only in October. The worst-case scenario was when the soil was only prepared in October, near sowing, highlighting the importance of early soil preparation to reduce imidazolinone residues.

Another important point addressed by Kraemer, Marchesan, Avila, Machado, Grohs, Massoni and Sartori [75] is the need to reduce injury in the crop's initial phase. The authors recommend remediating imidazolinones' residual effect by cultivating ryegrass (*Lolium multiflorum* L.) during the winter as a cover crop. Ryegrass is well adapted to floodplain soils and has reasonable tolerance to imidazolinone residues [11,76].

Other species have already been evaluated in terms of their effectiveness along these lines, mainly legumes, but their low adaptability to hydromorphic soils is a limitation.

However, the birdsfoot trefoil (*Lotus corniculatus* L.) and white clover (*Trifolium repens* L.) have been suggested as potential phytoremediators [77–79]. Further, Souto, et al. [80] demonstrated an increase in microbial activity in rhizosphere soil due to vetch (*Vicia sativa*) cultivation, while the cultivation of birdsfoot trefoil followed by white clover resulted in a 94–97% reduction in imidazolinone concentrations compared to the non-cultivated soil, making this an excellent option for imidazolinone remediation. The study also indicated that summer crops such as jack beans (*Canavalia ensiformis* (L.) DC.) and soybean have a similar effect in reducing imidazolinone residues.

The main advantage of growing soybean in rotation with rice is reducing the seed bank of weedy rice and other weeds. However, soybean cultivation also contributes to the degradation of imidazolinone residues due to the drainage and soil pH correction necessary for the crop to be established. Due to the self-liming effect of flooding, the pH correction process, generally, is a neglected technique for rice cultivation. According to Boeni, et al. [81], more than 75% of soils used in rice cultivation have a pH less than 5.5, limiting imidazolinones' availability for degradation. In contrast, a pH of 6.0 is necessary for soybean cultivation, promoting imidazolinone degradation (Figure 7).

In order to better manage imidazolinone residues in the soil, the results from this review indicated several management practices that can be adopted to reduce the carryover effect on subsequent crops. During the rice season, an alternative that may be used is the adoption of intermittent irrigation. This irrigation practice, besides maximizing the use of water, allows for soil aeration and, therefore, improves the rate of degradation of imidazolinones by aerobic microorganisms. However, there are few studies demonstrating that intermittent irrigation can be an efficient alternative to improve imidazolinone dissipation during the rice season. Therefore, it is suggested that this subject needs further investigation to better validate the technique for this purpose. After rice harvesting, soil drainage and the use of cover crops (such as ryegrass or other crops) allow for remediation and greater degradation (by aerobic microorganisms) of imidazolinone residues. As already mentioned, liming the field in the off-rice season would greatly benefit dissipation and improve soil quality to introduce other rotational crops, especially in the summer, such as soybean, for example. Therefore, rice farmers should consider these strategies isolated or jointly to minimize carryover problems. It is important to note that it is possible that the adoption of multiple strategies would result in a high dissipation effect. Some strategies are required to be used in combination such as drainage, liming, and rotation with soybean. However, results of the combined effect of these strategies are lacking in the current literature. Based on the discussion outlined in this review, the primary management strategies used to minimize carryover effects in lowland areas are summarized in Figure 12.

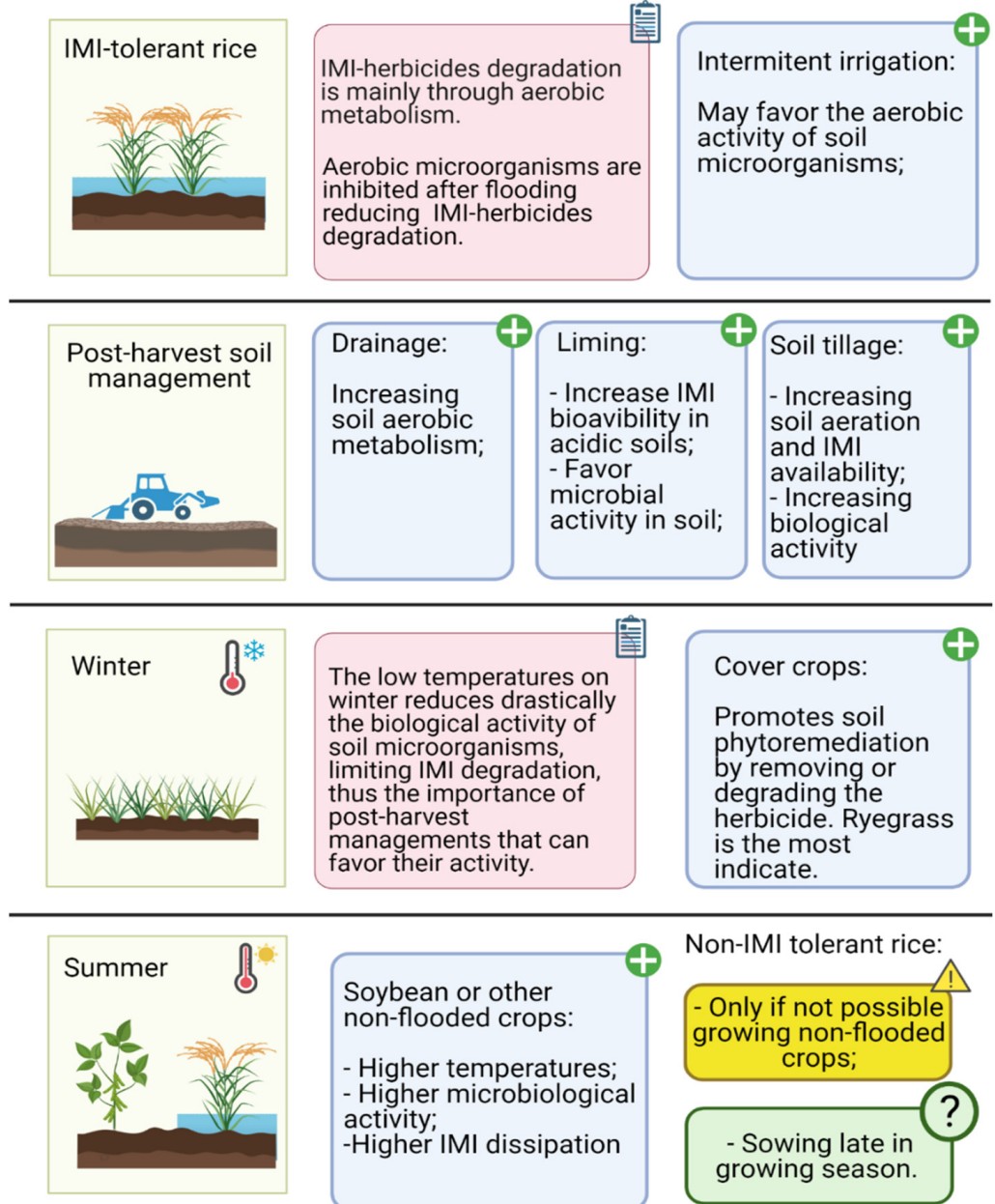

**Figure 12.** Summary of the recommendations to mitigate imidazolinone persistence in rice soils. From top to bottom is the timeline starting with the rice season using imidazolinone-resistant genotypes. The blue boxes indicate factors that favor the mitigation of the carryover effect; the red box indicates situations that limit the degradation of imidazolinones; the yellow box indicates alerting situations in which there may be an effect of carryover on subsequent crops; and the green box indicates a suggestion for which there may be reduction of the carryover effects in non-tolerant rice.

## 5. Final Remarks

The degradation rate is determined by edaphoclimatic factors that regulate the availability of the herbicide in the soil (texture, organic matter, pH, moisture, and temperature). Strategies for reducing imidazolinone residues should be primarily based on two pillars: increasing the herbicide availability in the soil and enhancing biological activity.

**Supplementary Materials:** The following are available online at https://www.mdpi.com/article/10.3390/agriculture11040299/s1, Figure S1: List of the ten most significant terms in each topic calculated with the β method. Topics were arbitrarily classified as environmental dynamics (1, 7 and 8), weed management (4 and 10), quantification methodologies (3), omics (5 and 9), and generic (2 and 6). Figure S2: Ordinary axis graph showing the selection of included/excluded records by topic. Topics were arbitrarily classified as environmental dynamics (1, 7 and 8), weed management (4 and 10), quantification methodologies (3), omics (5 and 9), and generic (2 and 6). Figure S3: Color guides of the bibliographic records of the data used throughout the text., and Dataset.

**Author Contributions:** Conceptualization, V.R.G., L.A.d.A. and E.R.C.; methodology, V.R.G.; formal analysis, V.R.G.; investigation, V.R.G., and E.R.C.; data curation, V.R.G., and E.R.C.; writing—original draft preparation, V.R.G., M.V.F., L.A.d.A., E.R.C.; writing—review and editing, V.R.G., M.V.F., L.A.d.A., E.R.C.; with approval from all authors; supervision, L.A.d.A., E.R.C.; project administration, L.A.d.A., E.R.C.; funding acquisition, L.A.d.A., E.R.C. All authors have read and agreed to the published version of the manuscript.

**Funding:** This research received external funding from CNPq (Conselho Nacional de Desenvolvimento Científico e Tecnológico) for the Research Fellowship of L.A.A./N.Proc. 310830/2019-2. This study was financed in part by the Coordenação de Aperfeiçoamento de Pessoal de Nível Superior-Brasil (CAPES)-Finance Code 001 by providing the student assistantship of V.R.G. and M.V.F.

**Institutional Review Board Statement:** Not applicable.

**Informed Consent Statement:** Not applicable.

**Acknowledgments:** The author thanks the Federal University of Pelotas for the staff support.

**Conflicts of Interest:** The authors declare no conflict of interest.

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
