# Peer review of "Understanding the Opportunities to Mitigate Carryover of Imidazolinone Herbicides in Lowland Rice"

_agriculture, doi:10.3390/agriculture11040299_

Round 1

Reviewer 1 Report

The manuscript "Understanding the opportunities to mitigate carryover of imidazolinones herbicides in lowland rice" by V.R. Gehrke et al. is devoted to a systematic review of the urgent information on the problem of imidazolinones herbicides for the development of management strategies to reduce imidazolinone persistence in the soil following the cultivation of tolerant rice. A systematic review carried out using the Prisma Protocol, allowed to interpret the scattered data and provided key recommendations for the implementation of the imidazolinone reduction strategy.

The manuscript is well and logically organized. However, the authors must make a number of clarifications in order for the manuscript to be accepted:

A complete description of the sources of information should be provided, namely: indicate the dates of coverage for the databases, as well as the date of the last search.

The "Final Remarks" section gives an average half-life interval from 40 to 100 days. However, this interval was not discussed in the manuscript.

It is necessary to uniformly indicate the titles of publications in the "References" section (see Ref. 7, 11-13, 18, 22, 24 and so on).

Author Response

Dear Editors,

Followed are the modifications incorporated in the manuscript draft.

Reviewer 1:

1- A complete description of the sources of information should be provided, namely: indicate the dates of coverage for the databases, as well as the date of the last search.

Answer: The information has been added to the text in Materials and Methods section.

2- The "Final Remarks" section gives an average half-life interval from 40 to 100 days. However, this interval was not discussed in the manuscript.

Answer: Sentence was removed from final remarks.

3- It is necessary to uniformly indicate the titles of publications in the "References" section (see Ref. 7, 11-13, 18, 22, 24 and so on).

Answer: The references were corrected and standardized.

ABSTRACT

Line 9-30: altered to “The Clearfield™ system (CL) is currently the primary tool for selectively weedy-rice management in irrigated rice. The application of imidazolinones herbicides and the use of tolerant cultivars allows effective and practical control to producers. However, their persistence in the soil for an extended period after harvesting causes damage to successor crops, in some cases limiting these areas to successive cultivation of imidazolinone non-tolerant genotypes. Thus, it is necessary to understand agricultural practices which can favor the dissipation of these herbicides. The objective of this systematic review was to analyze the factors that affect imidazolinones persistence on soil and use this information to provide management strategies to mitigate herbicide carryover in lowland rice. It was performed a literature review and the publications were selected using the determination of the half-life in the soil as selection criteria. The data were summarized according to the biotic conditions, soil parameters, and environmental variables after selecting the data. The data shows that the imidazolinone soil dissipation occurs primarily through biodegradation af-fected by soil conditions that allow greater bioavailability of the herbicide in the soil solution, e.g., higher pH, less clayey texture, moderate organic matter content, and high levels of soil moisture. Therefore, environmental conditions that favor biological activity, especially high temperatures, reduce the herbicide's half-life in the soil. Thus, the mitigation of imidazolinone soil persistence strategies should focus on improving herbicide availability and enhancing biological activity in the soil, especially in the rice off-season, where there are low temperatures that limit biodegra-dation. Cover and rotational crops, such as ryegrass and soybean are recommended, with the potential to mitigate soil residues. Establishment of crops other than rice would automatically enhance degradation rates as soil amendment practices such pH correction, and drainage practices would favor soil availability and biological activity.”

To

“(1) Background: The Clearfield™️ system (CL) is currently the primary tool for selectively weedy-rice management in irrigated rice. However, herbicide persistence in the soil may cause damage to successor crops. Thus, it is necessary to understand agricultural practices which can favor the dissipation of these herbicides. The objective of this study was to analyze the factors that affect imidazolinones' persistence and use this information to provide management strategies to mitigate carryover in lowland rice. (2) Methods: It was performed a literature review, and the publications were selected using the soil half-life parameter. The data were summarized according to the biotic conditions, soil parameters, and environmental variables. (3) Results: Imidazolinone dissipation in soil occurs primarily through biodegradation.  Herbicide biodegradation rate depends on environmental conditions such as temperature and bioavailability of the herbicide in the soil. Herbicide bioavailability is affected by soil conditions, with higher bioavailability in soil with higher pH, less clayey texture, moderate organic matter content, and higher soil moisture levels. Therefore, environmental conditions that favor biological activity, especially high temperatures, reduce the herbicide half-life in the soil. Strategies to mitigate carryover should focus on improving herbicide availability and enhancing biological activity in the soil, especially in the rice off-season, when low temperatures limit herbicide biodegradation. Cover and rotational crops, such as ryegrass and soybean, are recommended, with the potential to mitigate soil residues. (4) Conclusions: The establishment of crops other than rice would automatically enhance degradation rates as soil amendment practices such as pH correction and drainage practices would favor soil availability and biological activity.”

Introduction

Line 38-39: altered “the use of pesticides due to their practical and affordable nature and their” to “pesticide use due to its practical and affordable nature and its”

Line 48-50: altered “A particular case of herbicide carryover occurs with imidazolinone herbicides (imazapyr, imazapic, imazethapyr, imazamox, and imazaquin)” to “The imidazolinone herbicides (imazapyr, imazapic, imazethapyr, imazamox, and imazaquin) are, in general, soil persistent herbicides and can carryover affecting rotational crops”

Line 50: altered “high” to “specific”

Line 51-53: altered “which is why imazapyr is primarily in non-agricultural areas” to “An example of imazapyr which is used, is primarily in non-agricultural areas”

Line 57: altered “we call these cultivars as the” to “and were called”

Line 58: altered “cultivars and call them,” to “genotypes called”

Line 60: excluded “further”

Line 63-64: altered “Herbicide dynamics in these environments are completely different from in the uplands, mainly due to standing water during the growing season, which initially alters transport processes” to “Herbicide dynamics in these environments are remarkably different from uplands, mainly due to the maintenance of standing water during the growing season, which alters transport processes”

Line 66-67: altered “over affecting non-tolerant crops both in the winter and summer” to “that can affect non-tolerant crops both in the winter and summer (carryover)”

Line 69-70: altered “The result has been an increased reliance on herbicide-tolerant rice monoculture” to “As a consequence, there is a greater reliance on CL rice monoculture”

Line 74-75: altered “Herbicide half-life (t1/2), is the best estimate of herbicides persistence in the soil and is a good value to compare herbicides” to “Herbicide's half-life (t1/2) is the best estimate of herbicide persistence in the soil and is a valuable parameter to compare herbicides.”

Line 83: altered “following the cultivation of tolerant rice” to “after its use in the CL rice crop”

Materials and Methods

Line 85-86: altered “and searches were performed on the literature” to “Literature searches”

Line 89-91: added “The manuscripts were obtained from the databases cited between February 10, 2020 until March 30, 2020”

Line 91: altered “The records” to “The data obtained from this search”

Line 100-101: altered “(probability of a document being assigned to a topic)” to “(likelihood of a document being assigned to a subject)”

Results

Line 138: altered “records” to “documents”

Line 150: altered “evaluate” to “assess”

Line 161: altered “This grouping results” to “Results”

Line 164-167: altered “It should be noted that chemical compounds in the imidazolinone group can be broken down by abiotic means, as they are sensitive to photodegradation in water, depending on its turbidity, pH, and organic matter concentration (indirect photolysis)” to “Imidazolinone herbicide can be degraded by abiotic means as well, as they are sensitive to photodegradation in water, depending on its turbidity and soluble organic matter content (indirect photolysis)”

Line 174: excluded “factors such as”

Line 180: altered “the trend of soils” to “the soil trend”

Line 197-198: altered “matter reduces leads to a reduction in the degradation rate due to the greater sorption of the herbicide” to “matter reduces the degradation rate due to the herbicide's greater sorption”

Line 201: added “a”

Line 203: altered “effect” to “impact”

Line 203-206: altered “since the latter are weak acids, meaning that the carboxyl group's protonation leads the hydrophobic fraction of the soil to attract the herbicide at low pH” to “The imidazolinones are weak acids, meaning that the carboxyl group's protonation leads the soil's hydrophobic fraction to attract the herbicide's in low pH soils.”

Line 213: added “were”

Line 214: excluded “and the formation of 14CO2

Line 223-224: altered “consequently increases microbiological activity and successive degradation rates” to “increases microbiological activity and degradation rates”

Line 233: excluded “the increased amounts of”

Line 242-243: altered “In general, imidazolinone degradation is slower in conditions of low soil moisture due to lower availability” to “In general, imidazolinone degradation is slower in conditions of lower soil moisture”

Line 250-251: altered “noting a gradual increase in the degradation rate up to 75% and a reduction at 100% of field capacity, the latter of which was significant for imazethapyr” to “noted a gradual increase in the degradation rate up to 75% and a reduction at 100% of field capacity, especially for imazethapyr.”

Line 251-253: altered “These results highlight a gap in the literature, namely the degradation of imidazolinones in conditions of high moisture or soil saturation as in rice-producing areas, flooding in which results in a hypoxic environment” to “These results highlighted a gap in the literature considering the degradation of imidazolinones in conditions of higher moisture or soil saturation as in rice-producing areas, flooding results in a hypoxic environment”

Line 248: altered “assessed” to “assessing”

Line 258: excluded “simulated with nitrogen flow”

Line 263: altered “this” to “the”

Line 274-275: altered “this result” to "these results to”

Line 279: excluded “that”

Line 281: altered “allow for” to “will enable”

Line 283: altered “and temperature in that order” to “and secondly by temperature”

Line 291: altered “Another possibility addressed in the literature is” to “Another aspect that may affect degradation is”

Line 299: excluded “mixture”

Line 304: added “of”

Line 323-326: altered “According to the recommendations, these herbicides can be applied before emergence or just after emergence, followed by a second application in the V3-V4 stage of the rice plants soon before or simultaneously to flooding established” to “According to the recommendations, these herbicides can be applied before emergence or just after emergence, followed by a second application at the V3-V4 rice stage, before or simultaneously to flooding established”

Line 325: altered “Thus, the rice soil” to Rice soil”

Line 328: altered “been” to “being”

Line 334: altered “has been” to “as”

Line 339-343: altered “Figure 11. (A) Monthly average of minimum and maximum temperatures from 2000 to 2019 (Source: Dados da Rede do INMET [73]), (B) Average harvest and planting activity in Rio Grande do Sul between 2010 and 2019 (Source: IRGA [74]). The shaded area represents the percentage of Rio Grande do Sul State (biggest rice producer in Brazil) acreage with rice crop in progress from establishment (sowing) to harvest” to “Figure 11. (A) Monthly average of minimum and maximum temperatures from 2000 to 2019 (Source: Dados da Rede do INMET [73] ), (B) Average harvest and planting activity in Rio Grande do Sul state, Brazil between 2010 and 2019 (Source: IRGA [74]). The shaded area represents the percentage of Rio Grande do Sul State (biggest rice producer in Brazil) acreage with rice crop in progress from initial establishment (sowing starting in Sep) to final harvesting (end of May).”

Line 345: altered “to provide conditions to soil preparation” to “provide soil preparation conditions”

Line 348: excluded “as”

Line 349: added “preparation”

Line 354: added “the need to reduce”

Line 368-369: altered “rotation with rice is reducing the seedbank” to “rotation with rice is reducing the seed bank”

Line 376-378: added “Based on the discussion outlined in this review, the primary management strategies used to minimize carryover effects on lowland areas are summarized in Figure 12”

Figure 12 was added.

Final Remarks

Line 381: excluded “Imidazolinones can persist in soil for lengthy periods, with an average half-life of 40 to 100 days”

Line 388: altered “encouraging” to “enhancing”

We hope to have addressed all the comments. If there is anything else that needs to be explained/altered, please let us know.

Appreciate your suggestions and contributions with our manuscript.

Best regards,

Authors

Reviewer 2 Report

The study proposes a systematic literature review on the use of the imidazolinone herbicides, focusing on the degradation and half-life values.

The manuscript is well written and organized. However two major flaws can be found:

  • The review was carried out on scientific studies both in English and Portoguese: the choice of the latter is quite unusual since English is considered the international scientific language and inclusion of a different language might reduce the validity of the study, making it not replicable at an international level (see: doi: 13105/wjma.v7.i3.66). This point should be discussed and explained.
  • The research findings are unclear: while the description of the results is well detailed, the discussion of them is not sufficient. The Authors should highlight the research findings and their implications at a practical level. In doing this, the benefits of understanding the behaviour of herbicides should be discussed not only taking into account the environmental impact, but also the importance in terms of health and safety of farmers (e,g, you might consider the following studies: https://doi.org/10.1007/s12088-019-00841-x; ; https://doi.org/10.3390/ijerph16030310)

Additionally, the abstract should be modified following the journal suggestions: in particular, improving the Results (Summarize the article's main findings) and Conclusion(Indicate the main conclusions or interpretations) parts.

Author Response

Dear Editors,

Followed are the modifications incorporated in the manuscript draft.

Reviewer 2:

1- The review was carried out on scientific studies both in English and Portuguese: the choice of the latter is quite unusual since English is considered the international scientific language and inclusion of a different language might reduce the validity of the study, making it not replicable at an international level (see: doi: 13105/wjma.v7.i3.66). This point should be discussed and explained.

Answer: Clearfield system using a greater variety of imidazolinoes herbicides (imazethapyr, imazapyr and imazapic) is unique of some countries in South America, where Brazil (a Portuguese speaking country), is by far the primary player. Brazil is the only country outside Asia ranging among the top 10 rice producers. During the review it was observed that were a great number of relevance research conducted in Brazil. These researches are available (full text at least) in Portuguese. By these reasons, authors decided to included publication written in Portuguese to get a more robust understanding of the scenarios worldwide.

2- The research findings are unclear: while the description of the results is well detailed, the discussion of them is not sufficient. The Authors should highlight the research findings and their implications at a practical level. In doing this, the benefits of understanding the behaviour of herbicides should be discussed not only taking into account the environmental impact, but also the importance in terms of health and safety of farmers (e,g, you might consider the following studies: https://doi.org/10.1007/s12088-019-00841-x; ; https://doi.org/10.3390/ijerph16030310)

Answer: The practical results of the research were summarized and added in Figure 12 to encompass the review suggestion.

3- Additionally, the abstract should be modified following the journal suggestions: in particular, improving the Results (Summarize the article's main findings) and Conclusion (Indicate the main conclusions or interpretations) parts.

Answer: These adjustments were performed.

ABSTRACT

Line 9-30: altered “The Clearfield™ system (CL) is currently the primary tool for selectively weedy-rice management in irrigated rice. The application of imidazolinones herbicides and the use of tolerant cultivars allows effective and practical control to producers. However, their persistence in the soil for an extended period after harvesting causes damage to successor crops, in some cases limiting these areas to successive cultivation of imidazolinone non-tolerant genotypes. Thus, it is necessary to understand agricultural practices which can favor the dissipation of these herbicides. The objective of this systematic review was to analyze the factors that affect imidazolinones's persistence on soil and use this information to provide management strategies to mitigate herbicide carryover in lowland rice. It was performed a literature review and the publications were selected using the determination of the half-life in the soil as selection criteria. The data were summarized according to the biotic conditions, soil parameters, and environmental variables after selecting the data. The data shows that the imidazolinone soil dissipation occurs primarily through biodegradation af-fected by soil conditions that allow greater bioavailability of the herbicide in the soil solution, e.g., higher pH, less clayey texture, moderate organic matter content, and high levels of soil moisture. Therefore, environmental conditions that favor biological activity, especially high temperatures, reduce the herbicide's half-life in the soil. Thus, the mitigation of imidazolinone soil persistence strategies should focus on improving herbicide availability and enhancing biological activity in the soil, especially in the rice off-season, where there are low temperatures that limit biodegra-dation. Cover and rotational crops, such as ryegrass and soybean are recommended, with the potential to mitigate soil residues. Establishment of crops other than rice would automatically enhance degradation rates as soil amendment practices such pH correction, and drainage practices would favor soil availability and biological activity.”

To

“(1) Background: The Clearfield™️ system (CL) is currently the primary tool for selectively weedy-rice management in irrigated rice. However, herbicide persistence in the soil may cause damage to successor crops. Thus, it is necessary to understand agricultural practices which can favor the dissipation of these herbicides. The objective of this study was to analyze the factors that affect imidazolinones' persistence and use this information to provide management strategies to mitigate carryover in lowland rice. (2) Methods: It was performed a literature review, and the publications were selected using the soil half-life parameter. The data were summarized according to the biotic conditions, soil parameters, and environmental variables. (3) Results: Imidazolinone dissipation in soil occurs primarily through biodegradation.  Herbicide biodegradation rate depends on environmental conditions such as temperature and bioavailability of the herbicide in the soil. Herbicide bioavailability is affected by soil conditions, with higher bioavailability in soil with higher pH, less clayey texture, moderate organic matter content, and higher soil moisture levels. Therefore, environmental conditions that favor biological activity, especially high temperatures, reduce the herbicide half-life in the soil. Strategies to mitigate carryover should focus on improving herbicide availability and enhancing biological activity in the soil, especially in the rice off-season, when low temperatures limit herbicide biodegradation. Cover and rotational crops, such as ryegrass and soybean, are recommended, with the potential to mitigate soil residues. (4) Conclusions: The establishment of crops other than rice would automatically enhance degradation rates as soil amendment practices such as pH correction and drainage practices would favor soil availability and biological activity”

Introduction

Line 38-39: altered “the use of pesticides due to their practical and affordable nature and their” to “pesticide use due to its practical and affordable nature and its”

Line 48-50: altered “A particular case of herbicide carryover occurs with imidazolinone herbicides (imazapyr, imazapic, imazethapyr, imazamox, and imazaquin)” to “The imidazolinone herbicides (imazapyr, imazapic, imazethapyr, imazamox, and imazaquin) are, in general, soil persistent herbicides and can carryover affecting rotational crops”

Line 50: altered “high” to “specific”

Line 51-53: altered “which is why imazapyr is primarily in non-agricultural areas” to “An example of imazapyr which is used, is primarily in non-agricultural areas”

Line 57: altered “we call these cultivars as the” to “and were called”

Line 58: altered “cultivars and call them,” to “genotypes called”

Line 60: excluded “further”

Line 63-64: altered “Herbicide dynamics in these environments are completely different from in the uplands, mainly due to standing water during the growing season, which initially alters transport processes” to “Herbicide dynamics in these environments are remarkably different from uplands, mainly due to the maintenance of standing water during the growing season, which alters transport processes”

Line 66-67: altered “over affecting non-tolerant crops both in the winter and summer” to “that can affect non-tolerant crops both in the winter and summer (carryover)”

Line 69-70: altered “The result has been an increased reliance on herbicide-tolerant rice monoculture” to “As a consequence, there is a greater reliance on CL rice monoculture”

Line 74-75: altered “Herbicide half-life (t1/2), is the best estimate of herbicides persistence in the soil and is a good value to compare herbicides” to “Herbicide's half-life (t1/2) is the best estimate of herbicide persistence in the soil and is a valuable parameter to compare herbicides.”

Line 83: altered “following the cultivation of tolerant rice” to “after its use in the CL rice crop”

Materials and Methods

Line 85-86: altered “and searches were performed on the literature” to “Literature searches”

Line 89-91: added “The manuscripts were obtained from the databases cited between February 10, 2020 until March 30, 2020”

Line 91: altered “The records” to “The data obtained from this search”

Line 100-101: altered “(probability of a document being assigned to a topic)” to “(likelihood of a document being assigned to a subject)”

Results

Line 138: altered “records” to “documents”

Line 150: altered “evaluate” to “assess”

Line 161: altered “This grouping results” to “Results”

Line 164-167: altered “It should be noted that chemical compounds in the imidazolinone group can be broken down by abiotic means, as they are sensitive to photodegradation in water, depending on its turbidity, pH, and organic matter concentration (indirect photolysis)” to “Imidazolinone herbicide can be degraded by abiotic means as well, as they are sensitive to photodegradation in water, depending on its turbidity and soluble organic matter content (indirect photolysis)”

Line 174: excluded “factors such as”

Line 180: altered “the trend of soils” to “the soil trend”

Line 197-198: altered “matter reduces leads to a reduction in the degradation rate due to the greater sorption of the herbicide” to “matter reduces the degradation rate due to the herbicide's greater sorption”

Line 201: added “a”

Line 203: altered “effect” to “impact”

Line 203-206: altered “since the latter are weak acids, meaning that the carboxyl group's protonation leads the hydrophobic fraction of the soil to attract the herbicide at low pH” to “The imidazolinones are weak acids, meaning that the carboxyl group's protonation leads the soil's hydrophobic fraction to attract the herbicide's in low pH soils.”

Line 213: added “were”

Line 214: excluded “and the formation of 14CO2

Line 223-224: altered “consequently increases microbiological activity and successive degradation rates” to “increases microbiological activity and degradation rates”

Line 233: excluded “the increased amounts of”

Line 242-243: altered “In general, imidazolinone degradation is slower in conditions of low soil moisture due to lower availability” to “In general, imidazolinone degradation is slower in conditions of lower soil moisture”

Line 250-251: altered “noting a gradual increase in the degradation rate up to 75% and a reduction at 100% of field capacity, the latter of which was significant for imazethapyr” to “noted a gradual increase in the degradation rate up to 75% and a reduction at 100% of field capacity, especially for imazethapyr.”

Line 251-253: altered “These results highlight a gap in the literature, namely the degradation of imidazolinones in conditions of high moisture or soil saturation as in rice-producing areas, flooding in which results in a hypoxic environment” to “These results highlighted a gap in the literature considering the degradation of imidazolinones in conditions of higher moisture or soil saturation as in rice-producing areas, flooding results in a hypoxic environment”

Line 248: altered “assessed” to “assessing”

Line 258: excluded “simulated with nitrogen flow”

Line 263: altered “this” to “the”

Line 274-275: altered “this result” to "these results to”

Line 279: excluded “that”

Line 281: altered “allow for” to “will enable”

Line 283: altered “and temperature in that order” to “and secondly by temperature”

Line 291: altered “Another possibility addressed in the literature is” to “Another aspect that may affect degradation is”

Line 299: excluded “mixture”

Line 304: added “of”

Line 323-326: altered “According to the recommendations, these herbicides can be applied before emergence or just after emergence, followed by a second application in the V3-V4 stage of the rice plants soon before or simultaneously to flooding established” to “According to the recommendations, these herbicides can be applied before emergence or just after emergence, followed by a second application at the V3-V4 rice stage, before or simultaneously to flooding established”

Line 325: altered “Thus, the rice soil” to Rice soil”

Line 328: altered “been” to “being”

Line 334: altered “has been” to “as”

Line 339-343: altered “Figure 11. (A) Monthly average of minimum and maximum temperatures from 2000 to 2019 (Source: Dados da Rede do INMET [73]), (B) Average harvest and planting activity in Rio Grande do Sul between 2010 and 2019 (Source: IRGA [74]). The shaded area represents the percentage of Rio Grande do Sul State (biggest rice producer in Brazil) acreage with rice crop in progress from establishment (sowing) to harvest” to “Figure 11. (A) Monthly average of minimum and maximum temperatures from 2000 to 2019 (Source: Dados da Rede do INMET [73] ), (B) Average harvest and planting activity in Rio Grande do Sul state, Brazil between 2010 and 2019 (Source: IRGA [74]). The shaded area represents the percentage of Rio Grande do Sul State (biggest rice producer in Brazil) acreage with rice crop in progress from initial establishment (sowing starting in Sep) to final harvesting (end of May).”

Line 345: altered “to provide conditions to soil preparation” to “provide soil preparation conditions”

Line 348: excluded “as”

Line 349: added “preparation”

Line 354: added “the need to reduce”

Line 368-369: altered “rotation with rice is reducing the seedbank” to “rotation with rice is reducing the seed bank”

Line 376-378: added “Based on the discussion outlined in this review, the primary management strategies used to minimize carryover effects on lowland areas are summarized in Figure 12”

Figure 12 was added.

Final Remarks

Line 381: excluded “Imidazolinones can persist in soil for lengthy periods, with an average half-life of 40 to 100 days”

Line 388: altered “encouraging” to “enhancing”

We hope to have addressed all the comments. If there is anything else that needs to be explained/altered, please let us know.

Appreciate your suggestions and contributions with our manuscript.

Best regards,

Authors

Round 2

Reviewer 2 Report

The Authors have improved the manuscript considerably. However, in this reviewer’s opinion the criticality related to the discussion of results seems not solved.

Aa a matter of facts, although Figure 12 is “fascinating”, its contents need to be explained in detail, highlighting the meaning and scientific soundness of each one of its parts. Then, once again, it is suggested to pinpoint the research findings and their implications at a practical level. In doing this, the benefits of understanding the behaviour of herbicides should be discussed not only taking into account the environmental impact, but also the importance in terms of health and safety of farmers (e,g, you might consider the following studies: https://doi.org/10.1007/s12088-019-00841-x; ; https://doi.org/10.3390/ijerph16030310).

Actually, despite a good analysis of the extant literature, the Authors should provide a discussion of what can be derived by the literature review, taking into account all implications related to the herbicide use, both from the soil and the farmers’ standpoint. Based on this strategies to mitigate these effects should be suggested (i.e. explained) considering the literature outputs, and further research work should be addressed.

Given the quality and impact of the journal, in this reviewer’s opinion, it is necessary that the discussion section aims to integrate rather than just list the findings by different studies. To augment the quality of the study, direct interpretation of the findings of the study should be made and suggestion of implications for future research or practice.

Round 3

Reviewer 2 Report

The Authors have improved the manuscript sufficiently. Hence it can be considered for publication.